# Non-*Saccharomyces* Yeasts from Organic Vineyards as Spontaneous Fermentation Agents

**DOI:** 10.3390/foods12193644

**Published:** 2023-10-02

**Authors:** Lorena López-Enríquez, Josefina Vila-Crespo, José Manuel Rodríguez-Nogales, Encarnación Fernández-Fernández, Violeta Ruipérez

**Affiliations:** 1Microbiology Department, Higher Technical School of Agrarian Engineering of Palencia, University of Valladolid, Av. Madrid 50, 34004 Palencia, Spain; lorena.lopez.enriquez@estudiantes.uva.es (L.L.-E.); josefinamaria.vila@uva.es (J.V.-C.); 2Food Technology Department, Higher Technical School of Agrarian Engineering of Palencia, University of Valladolid, Av. Madrid 50, 34004 Palencia, Spain; josemanuel.rodriguez@uva.es (J.M.R.-N.); encarnacion.fernandez@uva.es (E.F.-F.)

**Keywords:** yeast ecology, microbial diversity, enzymatic activity, wine quality, Verdejo wine

## Abstract

Currently, non-*Saccharomyces* yeasts are the subject of interest, among other things, for their contribution to the aromatic complexity of wines. In this study, the characterisation of non-*Saccharomyces* yeasts was addressed by their isolation during spontaneous fermentations of organic Verdejo grapes, obtaining a total of 484 isolates, of which 11% were identified by molecular techniques as non-*Saccharomyces* yeasts. Fermentative isolates belonging to the species *Hanseniaspora meyeri*, *Hanseniaspora osmophila*, *Pichia guilliermondii*, *Pichia kudriavzevii*, *Torulaspora delbrueckii*, and *Wickerhamomyces anomalus* were analysed. Significant differences were found in the yeast populations established at the different fermentation stages. Interestingly, *W. anomalus* stood up as a widely distributed species in vineyards, vintages, and fermentation stages. Several of the strains studied stood out for their biotechnological potential in the production of Verdejo wine, showing the presence of relevant enzymatic activity for the release of varietal aromas and the technological improvement of the winemaking process. Three enzymatic activities were found in an important number of isolates, β-glucosidase, protease, and β-lyase, implicated in the positive aromatic impact on this style of white wine. In that sense, all the isolates of *W. anomalus* presented those activities. *T. delbrueckii* isolates were highlighted for their significant β-lyase activity. In addition, *T. delbrueckii* was outlined because of its potential to achieve an elevated fermenting power, as well as the lack of lag phase. The results obtained highlight the importance of maintaining the microbial diversity that contributes to the production of wines with unique and distinctive characteristics of the production region.

## 1. Introduction

The spontaneous alcoholic fermentation is carried out by a succession of yeasts of different species present in the grape must, comprising a mixed and sequential participation of non-*Saccharomyces* and *Saccharomyces* yeasts. Although non-*Saccharomyces* yeasts have been considered undesirable spoilage microorganisms in the production of wine for decades, nowadays, they are considered to confer authenticity and enhance regional characteristics of wines [1,2,3].

The relevance of indigenous strains in the winemaking process has been widely reported. Regarding non-*Saccharomyces* yeasts, specific technological properties and quality parameters, such as the production of certain enzymes involved in the distinctive aromatic profiles, among others, are becoming important to obtain high-quality wines [4,5,6,7]. Several species of non-*Saccharomyces* yeasts, some of which are commercially available, have been described to participate in alcoholic fermentation, contributing to generating aromatic compounds and modifying metabolites of the final product [8,9,10,11,12].

Tendencies in winemaking focused on responding to consumers’ demand for wines with regional distinctive characteristics need to be linked to ensuring a complete and efficient alcoholic fermentation. In this sense, some winemakers consider the need for spontaneous fermentations to obtain original wines reflecting the terroir in each area [3]. Despite being a cheap alternative, limitations related to an inadequate fermentation kinetic, lack of microbial control, or stuck fermentation often result in a certain reluctance to develop spontaneous fermentation among producers. To solve this issue, several strategies have been proposed, such as using non-*Saccharomyces* starters combined with *S. cerevisiae*, multi-strain starters, or pied de cuve to perform alcoholic fermentation [3]. However, all the strategies proposed require exhaustive ecological studies to be able to predict and control the microbiota implicated in the organoleptic profile of the final product.

Recent studies underline the relevance of understanding the mechanisms in wine ecosystems, evaluating the impact of individual yeast species on the dynamic of the microbial population and on the aromatic properties of wine, as well as describing the yeast interactions during wine fermentation [13,14,15,16]. The analysis of the microbial biodiversity from different vineyards and viticultural areas demonstrates that grape and wine microbiota influence regional patterns of wine, providing evidence that microbial activity is associated with wine terroir [17,18]. Thus, there is still a significant gap in knowledge on yeast interactions in the grape–wine ecosystem, pointing at ecological studies as an essential step to understanding the dynamic of the microbiota in the winemaking process and improving the quality of wines through oenological practices.

White wine, elaborated with Verdejo grape, the main variety of the Appellation of Origin (AO) Rueda (North Central Spain), is one of the most important Spanish white wines and a significant driving force of Rueda’s region economy. Despite being a unique and high-quality wine, there is a lack of microbial ecological studies to elucidate the indigenous yeasts that play a part in its regional distinctiveness. Previous studies in our group focused on the diversity of the *S. cerevisiae* population in spontaneous fermentation of Verdejo wine, highlighting the effect of vineyard and vintage on yeast communities as well as the presence of singular strains for each of the populations analysed [19]. 

To take a step forward, the aim of this study was to analyse non-*Saccharomyces* isolates obtained from those spontaneous fermentations. For this purpose, non-*Saccharomyces* species from spontaneous fermentations in the winery of grapes coming from three different organic vineyards during two vintages were identified using molecular techniques. Their oenological characterisation was carried out through the determination of kinetic parameters and relevant enzymatic activities in the production of Verdejo wine. 

## 2. Materials and Methods

### 2.1. Isolation of Yeasts from Spontaneous Fermentations

Organic Verdejo grapes of three separated vineyards (V1, V2 and V3) located in the AO Rueda were harvested in vintages 2010 and 2012 (first and second). The work in the vineyards, bellowing to Belondrade winery (La Seca, Valladolid, Spain), was carried out practising organic viticulture, which uses neither herbicides nor pesticides. This encourages the existing biodiversity in the vineyard. A strict selection of grapes was carried out in the vineyard and later on the sorting table in the winery, obtaining high-quality grapes for the winemaking process. 

After grapes were destemmed and crushed, 4 g/hL of total sulphur dioxide was added. Spontaneous fermentative processes related to each combination of vineyard vintage were developed in 300 L oak barrels in the winery; sampling was carried out at different stages of the winemaking process for the isolation of yeasts: freshly crushed grape must, CM; racked must, RM; start of fermentation, SF; tumultuous fermentation, TF; end of fermentation, EF. Samples were diluted and spread onto plates of YPD medium, containing 1% (*w*/*v*) yeast extract (Biolife, Milano, Italy), 2% (*w*/*v*) peptone (Panreac, Barcelona, Spain), 2% (*w*/*v*) dextrose (Scharlab, Barcelona, Spain), 2% (*w*/*v*) agar (Scharlab). The agar plates were incubated at 25 °C for 5 days. At each fermentation point, yeast isolates obtained as separated colonies were picked up and subsequently analysed. A total of 484 isolates were analysed, establishing 54 different genetic groups of *S. cerevisiae* that comprised 89% of the isolates [19] and 11% of non-*Saccharomyces* species (Appendix A).

### 2.2. Molecular Identification of the Isolates

The isolates were allowed to grow for 18–36 h in 1.5 mL of YPD broth at 25 °C with shaking (220 rpm), and yeast genomic DNA was isolated according to the protocol previously described [20].

Yeast isolates were classified into different molecular groups based on digestion patterns obtained by restriction fragment length polymorphism (RFLP) of the 5.8S-ITS region of ribosomal DNA (rDNA). Primers ITS1 (5′-TCC GTA GGT GAA CCT GCG G-3′) and ITS4 (5′-TCC TCC GCC GCT TAT TGA TAT GC-3′) (Sigma-Aldrich, San Luis, MO, USA) were used to amplify the rDNA region [21]. Then, the PCR product was digested by using the restriction enzymes *HaeIII*, *HinfI* and *CfoI* (10 U/µL; Fisher Scientific, Madrid, Spain), following the manufacturer’s recommendations. Both the amplified fragment and their three digestion products were separated in 4.5% (*w*/*v*) D1 Low EEO agarose gels (Pronadisa, Madrid, Spain) in TAE 1X (Fisher Scientific), applying a current of 120 V for 3 h, and running a distance of approximately 5 cm. The electrophoresis progress was determined by using the molecular weight marker GeneRuler 100 bp DNA Ladder (Fisher Scientific) in all electrophoresis assays. Molecular profiles were visualized in Gel Doc XR+ gel documentation system (BioRad, Hercules, CA, USA) after post-electrophoresis staining with GelRed (Biotium, Fremont, CA, USA).

Representative isolates of each molecular group obtained through the RFLP method were analysed by sequencing the D1/D2 region of the 28S RNA gene to confirm their molecular identification at the species level.

### 2.3. Enzymatic Activities

#### 2.3.1. β-Glucosidase Activity

β-glucosidase activity was evaluated on a medium containing 0.5% (*w*/*v*) arbutine (Sigma-Aldrich), 0.1% (*w*/*v*) yeast extract (Labkem, Dublin, Ireland), and 2% (*w*/*v*) agar (Difco, Detroit, MI, USA). The components were dissolved in distilled water, and 2.0 mL of a 1% (*w*/*v*) iron chloride (Panreac) solution was added for each 100 mL [22]. The medium was autoclaved at 121 °C for 15 min before adding to the plates. A single colony was spread onto the surface, and the plates were incubated at 26 °C for 15 days. Dark-black cultures were considered positive. 

#### 2.3.2. Protease Activity

Protease activity was determined on YPD medium (1% (*w*/*v*) yeast extract (Labkem), 2% (*w*/*v*) peptone (Panreac), 2% (*w*/*v*) dextrose (Labkem), and 2% (*w*/*v*) agar (Difco)) containing 2% (*w*/*v*) skim milk powder [23]. YPD medium and skim milk were autoclaved at 121 °C for 15 min separately and mixed before adding to Petri dishes. A single colony was spread onto the plates and was incubated at 26 °C for 5–7 days. A clear halo around the colonies was considered a positive protease activity. 

#### 2.3.3. β-Glucanase Activity

β-glucanase activity was determined on a YPD medium containing 0.2% (*w*/*v*) yeast β-glucan (Megazyme, Neogen, Lansing, MI, USA) [5]. The medium was autoclaved at 121 °C for 15 min and poured into Petri dishes. A single colony was spread onto the plates, and the plates were incubated at 25 °C for 5–7 days. Afterwards, the colonies were rinsed off with distilled water, and the surfaces of the plates were covered with 0.03% (*w*/*v*) Congo red solution (Sigma-Aldrich) for 15 min. Positive activity was confirmed as a clear halo on the surface appeared where the colonies had grown.

#### 2.3.4. β-Lyase Activity

β-lyase activity was evaluated with a culture medium containing 1.2% (*w*/*v*) yeast carbon base (Difco), 0.1% (*w*/*v*) S-methyl-L-cysteine (Panreac), 0.01% (*w*/*v*) pyridoxal-5′-phosphate (Panreac), and 2% agar (Difco). The agar solution was autoclaved at 121 °C for 15 min, and all other components were sterilized by filtration (0.22 µm) after adjusting the pH to 3.5 with a 1 M HCl solution. Finally, solutions were mixed and poured into Petri dishes [23]. A single colony was spread onto the plate surface and incubated at 26 °C for 48 h. Afterwards, if growth was observed, a single colony from this plate was spread onto another plate. Enzymatic activity was considered positive when the growth of the isolates was significant in both plates.

In order to estimate differences in the growth of the positive β-lyase yeasts analysed, the same medium was prepared without agar. Colonies coming from plates containing the solid medium described above were resuspended in peptone water (1.0 g/L peptone, 8.5 g/L NaCl, pH 7.0 ± 0.2), and 10^6^ CFU/mL were inoculated in 1 mL of the liquid medium. Growth was estimated by optical density at 600 nm after 48 h of incubation at 26 °C. All the experiments were carried out in triplicate.

### 2.4. Fermentation Conditions

Microfermentations were carried out in a 100-mL Erlenmeyer flask containing 50 mL of Verdejo must autoclaved at 121 °C for 15 min. At this point, sugar content and pH were determined (22.3 ± 0.1 °Brix; pH 3.48 ± 0.20). The yeasts were cultured in YPD broth overnight, and cell proliferation was determined spectrophotometrically by measuring the optical density at 600 nm. The flasks were inoculated with 10^6^ CFU/mL of the non-*Saccharomyces* yeasts, sealed with a fermentation cap and incubated at 21 °C. The loss in weight using an analytical balance was followed to estimate the CO_2_ production until the end of fermentation. All the experiments were carried out in triplicate. Kinetic curves were fitted using DMFit web edition (Institute of Food Research, Norwich, UK, http://www.dmfit.com/eng/, accessed on 7 September 2023) [24], and the potential maximum rate (µ_max_), lag phase (lag), maximum CO_2_ production (yEnd), and fermenting power (FP) were determined. 

### 2.5. Statistical Analysis

Yeast ecology studies involved the determination of α-diversity and β-diversity in the succession of yeast communities through the fermentation process. Species relative abundance ($p_{i}$, number of isolates of the same species) and richness (S, number of different species) revealed the diversity within the yeast populations (α-diversity) by calculating Shannon index ($H^{{}^{\prime }}=-\sum_{i=1}^{S}p_{i}\ln\left(p_{i}\right)$) [25,26]. Differences between the yeast populations considering vineyard (V1, V2 and V3), vintage (first and second), and fermentation stage (CM, RM, SF, TF, EF) as variables were determined by analysis of variance (ANOVA) on the Shannon index values obtained. Tukey’s test was used when the differences established were significant (*p*-value < 0.05). 

The dissimilarity between yeast populations (β-diversity) was established by non-metric multidimensional scaling (NMDS). NMDS algorithm was applied on a Bray–Curtis dissimilarity matrix computed from the abundance data matrix obtained according to the variables indicated previously for ANOVA [27,28].

Based on the enzymatic activities determined, different isolate groups were defined by cluster analysis. The similarity among the enzymatic behaviour of the strains was calculated by Ward’s method based on a squared Euclidean distances matrix and was represented by a dendrogram. Differences in β-lyase activity were computed by ANOVA, and isolates were considered as a whole or taking into account the species as a variable.

Categorical Principal Component Analysis (CATPCA) showed the relationships between objects (isolates) and variables analysed (ecological factors: vineyard, vintage, fermentation stage and species; fermentation features: FP, µ_max_, yEnd and lag; enzymatic activities: β-glucosidase, β-glucanase, β-lyase, and protease).

All the statistical analyses were carried out using the programs IBM SPSS Statistics version 26.0 (IBM Corp. in Armok, NY, USA) and Statgraphics Centurion version 19 (Statgraphics Technologies, Inc., The Plains, VA, USA). Ecological diversity indexes and distance matrices were calculated utilizing the Paleontological Statistics (PAST) software version 4.05 (Natural History Museum—University of Oslo, Norway) [29].

## 3. Results

### 3.1. Analysis of Biodiversity of Non-Saccharomyces Populations

Molecular techniques applied to the 484 yeast isolates obtained in the different spontaneous fermentation processes allowed for the identification of 55 non-*Saccharomyces* isolates (11% of the total isolates), confirming the presence of five genera and 10 species of different yeasts: *Hanseniaspora meyeri* (Hm); *Hanseniaspora osmophila* (Ho); *Papiliotrema laurentii* (Pl); *Papiliotrema terrestris* (Pt); *Pichia guilliermondii* (Pg); *Torulaspora delbrueckii* (Td); *Rhodotorula mucilaginosa* (Rm); *Naganishia globosa* (Ng); *Pichia kudriavzevii* (Pk); and *Wickerhamomyces anomalus* (Wa) (Figure 1). 

According to taxonomic studies [30], the isolated species *P. laurentii*, *P. terrestris*, and *N. globosa*, formerly *Cryptococcus* spp., as well as *R. mucilaginosa*, are unable to ferment wine sugars; therefore, they were classified as non-fermentative yeasts. These species were found exclusively in the first vintage and only in freshly crushed must (CM) (Figure 2). Species that exhibited fermentative capacity to some extent were identified as *H. meyeri*, *H. osmophila*, *P. guilliermondii*, *T. delbrueckii*, *P. kudriavzevii* and *W. anomalus*. On the one hand, some of these species were only found in the CM stage (*H. osmophila* and *P. kudriavzevii*). On the other hand, *W. anomalus* stood out as a widely distributed yeast, and it was found in all vineyards, vintages, and fermentation stages (Figure 2). The presence of *W. anomalus* isolates at SF, TF, and EF stages was a remarkable finding, as the dominant role of *S. cerevisiae* strains during the alcoholic fermentation process and the increasing alcohol concentrations, among other factors, adversely affected the survival of non-*Saccharomyces* yeasts [31]. Although the role of low-fermentative non-*Saccharomyces* yeasts in the aromatic fingerprinting of the wine is mainly limited to the initial stages of fermentation, some strains of *W. anomalus* have been recently described as ethanol tolerant (up to 12.5% *v*/*v*), which supports its wide distribution across the fermentation stages [2].

Based on the results of the distribution obtained, the ecological diversity of the yeast communities associated with vineyards, vintages, and fermentation stages was determined within populations (α-diversity) and among populations (β-diversity) (Figure 3). The number of isolates of each particular species in relation to the total number of isolates (relative abundance) and the number of different species (richness) at a given time allowed for computing the Shannon index. This ecological value defined the α-diversity of the populations of yeasts and enabled comparisons among them. Shannon index values ranged from 0.22 to 0.86 for the non-*Saccharomyces* populations defined in each vineyard and vintage considered, which indicated that yeast communities were composed of a few yeast species comprising a reduced number of isolates. No significant differences could be established in the Shannon index values of these yeast communities, showing a similar structure in terms of species abundance and richness. However, significant differences were demonstrated in the yeast populations established at the different fermentation stages. CM stage showed the highest Shannon index value (1.36), revealing significant differences in all fermentation stages (Figure 3A). These differences were associated with the exclusive presence of fermentative and non-fermentative yeasts at this stage. Based on our results, vineyard origin or vintage conditions did not affect the initial population structure. However, the concentration of non-fermentative yeasts and several fermentative yeasts in the early stages of fermentation has been reported to result in higher species diversity, showing the contribution of yeast from the soil, grapevine phyllosphere, and winery’s environment [32]. In addition, microbial ecology in grapes can be affected by a large number of factors, among which health status stands out, increasing the number and diversity of species [33]. Climate factors, such as rainfall, wind or temperature, may induce berry damage, affecting microbial communities. It has been previously reported that rainy vintages may increase damage to berries, increasing some species such as oxidative yeasts; however, the effect on microbiota is often not clear, as applying scientific methods is not easy. In our study, the meteorological conditions registered in both vintages [19] showed lower accumulated precipitation during the year in the second vintage; however, the volume of precipitation at harvest months was higher than in the first vintage. According to this data, it is not easy to establish a relationship between climate conditions and microbial diversity described in this study. Moreover, the exhaustive grape selection carried out by the winery during the harvest and entry of grapes minimized the presence of damaged berries in the winemaking process and did not allow for assessing their effect on microbial diversity of the fermentation processes. Although the concept of microbial terroir, linking microbial communities to a specific region or vineyard, has been subjected to contradictory studies, the requirement of incoming research to support or refute the concept is necessary [34].

Dissimilarity among yeast populations (β-diversity) was represented by an NMDS two-dimensional plot (Figure 3B). This mapping method shows the composition and distribution of each yeast community in a two-dimensional ordination space and highlights species or isolates that are shared between vineyards (V1, V2, V3) and vintages (first, second), such as those that are unique to a given time. Isolates identified in the first vintage were mainly plotted in the positive part of dimensions 1 and 2, while isolates from the second vintage were mainly mapped in the negative part of dimension 2. For vineyards, V2 was concentrated in the quadrant defined by the positive part of dimension 1 and the negative part of dimension 2. Taking into account the lower number of isolates detected in V1 and V2 in the first vintage, the clustering of isolates was close to the axis of the positive part of dimension 1. The isolates related to V3 showed more influence of the vintage factor due to the highest number of isolates in both vintages. The NMDS model obtained was supported by a very low-stress value of 0.00012.

The relationship between the microbial populations identified in the must and the organoleptic profile of the wine has been previously described by Bokulich et al. (2016), defining microbial composition as a biological marker that could drive and enhance the development of the fermentation process [17]. In this sense, the β-diversity represented by NMDS showed the oenological potential of each vineyard and the vintage based on the composition of their yeast populations. The diversity study was carried out considering all the isolates, as interesting oenological characteristics of yeasts have been previously defined as being species and strain-dependent [2,35]. 

### 3.2. Enzymatic Profile of Yeast Population

Species with potential oenological interest were included in subsequent technological characterisation. In that sense, non-fermentative yeasts were discarded, focusing the attention on those isolates that may be able to exhibit fermentative capacity to a greater or lesser extent. A total of 36 isolates of the genera *Hanseniaspora*, *Pichia*, *Torulaspora*, and *Wickerhamomyces* were evaluated in order to determine their enzymatic profile implicated in the modulation of the final organoleptic characteristics of wines.

Four enzymatic activities implicated in the organoleptic profile of Verdejo wine (β-lyase, β-glucosidase, β-glucanase, and protease) were chosen for their influence on the release of thiols and terpenes in the wine (β-lyase and β-glucosidase) and for their impact on fining processes (β-glucanase and protease) in the winery. Culture assays were developed using selective and differential media that showed remarkable growth or changes when the enzymatic activity was positive (Appendix A). The experimental approach based on culture-dependent techniques to determine enzymatic activities supposes a rapid and easy microbiological tool as the first step for the screening of yeast isolates [5,23]. Three enzymatic activities stood out due to the important number of positive isolates determined. β-glucosidase and protease activities were positive in 50% of the isolates, and β-lyase activity was positive in 100% of the isolates analysed. However, the percentage of isolates with β-glucanase activity was more moderate (11.1%) (Figure 4). According to our data, previous research demonstrated that β-glucosidase, protease, and β-lyase had widely distributed activities in different species of non-*Saccharomyces* yeasts [23].

Taking into account the non-*Saccharomyces* species identified, the protease activity was positive in all the isolates of the *W. anomalus* species (Wa01 to Wa16) and in the isolates from the *H. meye*ri (Hm01) and *H. osmophila* (Ho01) species. Similarly, β-glucosidase activity was determined in all isolates of *W. anomalus* species and in the isolates of the *P. guilliermondii* species (Pg01 and Pg02). In contrast, β-glucanase activity was revealed in a few isolates (Ho01, Wa03, Wa04, Wa05), and it was not related to a specific species. All the isolates presented β-lyase activity, being the unique enzymatic activity shown in every isolate of *P. kudriavzevii* (Pk01 to Pk13) and *T. delbrueckii* (Td01 to Td03). These results indicate a variable enzymatic behaviour—on the one hand, associated with certain non-*Saccharomyces* species and, on the other hand, related to a particular isolate within a wine yeast species. The contribution to wine aromatic complexity of two β-glucosidase positive strains (*Meyerozyma guilliermondii* NM218 and *Hanseniaspora uvarum* BF345) by sequential inoculation fermentation with two *S. cerevisiae* strains has been previously reported [36], evidencing that the differences in the aromatic profiles obtained were related to the specificity of the strains for the non-aromatic precursors of the must. Although similarities with other studies can be established, specific features found in the isolates of this study highlight the importance of understanding the diversity and evolution of the characteristic yeast populations involved in the production of a singular wine. 

Based on the results for the β-lyase activity, a new assay was proposed to evaluate it quantitatively in order to estimate differences within species. Six yeast isolates with significant β-lyase activity compared to the average activity calculated for all yeast isolates were pointed out: the isolates of species *T. delbrueckii* (Td01, Td02, Td03); *H. osmophila* (Ho); *H. meyeri* (Hm); and the isolate Wa12 from the species *W. anomalus*. Td03 was the isolate with the highest β-lyase activity of all isolates tested (Figure 5). Furthermore, significant variability was demonstrated in the behaviour of the isolates within species by ANOVA analysis (Appendix A). In this sense, the differences found between *W. anomalus* and *P. kudriavzevii* revealed the variability in the enzymatic behaviour within the same species. These results were in agreement with those demonstrated by Belda et al. (2016), where β-lyase was widespread in the isolates studied, and all *T. delbrueckii* isolates analysed had positive activity, although the enzymatic activity was moderate [35]. In our work, all three *T. delbrueckii* isolates were highlighted for their significant β-lyase activity, showing a great biotechnological potential both in the management of indigenous ferments in spontaneous fermentations of Verdejo grapes and in the possibility of selecting autochthonous strains to drive the fermentation towards a specific aromatic profile.

### 3.3. Fermentation Kinetics of Non-Saccharomyces Isolates

Fermentation kinetics under laboratory conditions were surveyed in different isolates of fermentative species. Fermentation curves were fitted to a sigmoid function [24]. Although some isolates achieved a high alcoholic degree (Table 1), none of the isolates were able to complete alcoholic fermentation, finding residual sugars (>28 g/L) in all the fermentations. Previous studies with indigenous non-*Saccharomyces* yeasts were selected because their enzymatic profiles also confirmed the inability to obtain dry wines, outlining the requirement of mixed fermentation with *S. cerevisiae* strains to achieve a total consumption of sugar [4,13,37]. 

On the one hand, *H. osmophila* and *T. delbrueckii* stood up for the high fermenting power (FP) and maximum specific rate of CO_2_ production (µ_max_), as well as the lack of lag phase (lag). Although *H. meyeri* also lacked a lag phase, the specific rate and fermenting power were moderate. In general, *Hanseniaspor*a spp. present a low fermenting power but are implicated in the production of interesting volatile compounds in the wine [1]. No significant differences were found in fermentative behaviour among the isolates of *T. delbrueckii*. The high fermentative potential of *T. delbrueckii* has been previously reported, placing this species as a suitable option when mixed fermentations with *S. cerevisiae* are considered [15]. On the other hand, *P. guilliermondi*i showed a scarce fermenting power coupled with a low maximum rate and a delayed beginning of fermentation (Table 1). Interestingly, *P. guilliermondii* has been reported to reduce final ethanol content in wine due to the aerobic metabolism of Crabtree-negative non-*Saccharomyces* yeasts [38] that supports the lack of CO_2_ production resulting in the prolonged lag phase observed in the present study.

Focusing our attention on the predominant species *P. kudriazevii* and *W. anomalus* (Table 1), both species showed a moderate maximum rate of CO_2_ production and no significant differences were found among the isolates within species. *P. kudriazevii* showed a moderate fermenting power, ranging from 4.3 to 9.5% v/v, finding variability in fermentation kinetics among the isolates of this species. The lag phase was detected in most of the isolates, with the exception of Pk07, Pk08, Pk09, and Pk11, which exhibited a prompt onset of fermentation. The presence of a lag phase may influence the fermentation time, as sequential fermentation with *S. cerevisiae* when non-*Saccharomyces* yeasts are present requires a longer period of time to finish the fermentation [39]. In contrast, *W. anomalus* isolates showed a low fermenting power, establishing significant differences between the lowest production of CO_2_ (Wa15) and the highest (Wa08 and Wa10). Although all the isolates exhibited a moderate lag phase and low maximum specific CO_2_ rate, no significant differences were found.

### 3.4. Screening of Oenological Traits

In order to have a complete overview of the global oenological relevance of the isolates that contribute to spontaneous fermentation, CATPCA analysis was applied to all the parameters analysed in this study. CATPC1 and CATPC2 displayed 48.39% and 15.82% of the variance, respectively, explaining the relationship between sets of all variables and isolates (Figure 6). On the one hand, the isolates sited in the positive values of CATPC1 were characterised by a medium–high fermenting power, in contrast to those situated in the negative part of the axis. On the other hand, CATPC2 allowed for the separation of widely distributed species either across vineyards or fermentation stages found within positive values from those found in a punctual geographical area, vintage, or stage of the winemaking process. 

Taken together, these results indicate that *W. anomalus* was a widely distributed species among vineyards and fermentation stages characterised mainly by protease and β-glucosidase activities as well as a low fermenting capacity. Some *W. anomalus* isolates (Wa04, Wa03 and Wa05) also stood out due to presenting β-glucanase activity. All *T. delbrueckii* isolates exhibited a high fermenting power as well as a high maximum specific rate of CO_2_ production and elevated β-lyase activity. Regarding *P. kudriavzevii*, this species was mainly defined by a vintage variability and moderate fermenting power and highlights the poor enzymatic activity displayed. In contrast, *P. guilliermondii* was defined by a delayed lag phase and low fermenting power. Considering the differences found in their oenological traits, spontaneous fermentation applies an adequate strategy to keep the contribution of indigenous strains to the final profile of Verdejo wine. 

## 4. Conclusions

Despite the relevance of the Verdejo grape variety in the wine sector, there is a lack of knowledge of the importance of indigenous yeasts in maintaining their regional distinctiveness and singularity. Ecological studies are an essential step to becoming aware of the importance of preserving indigenous microbiota that confers to the wine the organoleptic qualities demanded by consumers. In the present study, a variable distribution of non-*Saccharomyces* species across vineyards and vintages has been found, pointing to a high influence of environmental conditions on microbial biodiversity. Significant differences were found in yeast populations established at different fermentation stages. Interestingly, *W. anomalus* stood out as a widely distributed species in all vineyards, vintages, and fermentation stages. In addition, *T. delbrueckii* was outlined because of its potential to achieve an elevated fermenting power, as well as the lack of lag phase. Regarding the enzymatic activity of the isolates, several of the strains stood out for their biotechnological potential, showing the presence of relevant enzymatic activity for the release of varietal aromas and the technological improvement of the winemaking process. Three enzymatic activities were found in an important number of isolates, β-glucosidase, protease, and β-lyase, implicated in positive aromatic impact on this style of white wine. In that sense, all the isolates of *W. anomalus* presented those activities. These results highlight the importance of isolating and characterising indigenous non-*Saccharomyces* yeasts and open the possibility of identifying interesting strains to be used in mixed cultures. Further work under real winemaking conditions is now required to confirm the capability of the strains that showed suitable oenological properties in this study for modulating the final organoleptic profile of Verdejo wine.

## Figures and Tables

**Figure 1 foods-12-03644-f001:**
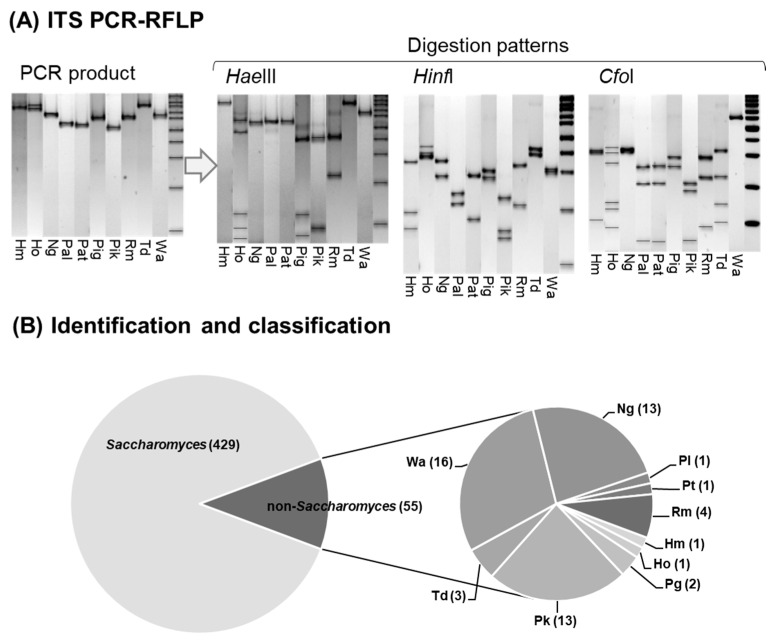
Identification of non-*Saccharomyces* isolates. (**A**) Molecular patterns obtained for each different yeast species by PCR-RFLP of the 5.8S-ITS region using ITS1 and ITS4 primers and restriction enzymes: *HaeIII*; *HinfI;* and *CfoI*. (**B**) Non-*Saccharomyces* yeast species and number of isolates (abundance) of each species from the initial collection of *Saccharomyces* and non-*Saccharomyces* yeasts were identified. *Hanseniaspora meyeri* (Hm), *Hanseniaspora osmophila* (Ho), *Papiliotrema laurentii* (Pl)*, Papiliotrema terrestris* (Pt), *Pich*ia *guilliermondii* (Pg), *Torulaspora delbrueckii* (Td), *Rhodotorula mucilaginosa* (Rm), *Naganishia globosa* (Ng), *Pichia kudriavzevii* (Pk), and *Wickerhamomyces anomalus* (Wa).

**Figure 2 foods-12-03644-f002:**
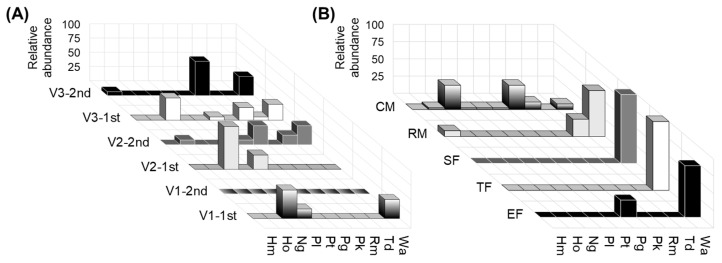
Distribution across vineyard, vintage, and fermentation stages. (**A**) Distribution of non-*Saccharomyces* across the vineyards (V1, V2, V3) and the first and second vintage (1st, 2nd). (**B**) Distribution of non-*Saccharomyces* across the fermentation stages (freshly crushed grape must, CM; racked must, RM; start of fermentation, SF; tumultuous fermentation, TF; end of fermentation, EF).

**Figure 3 foods-12-03644-f003:**
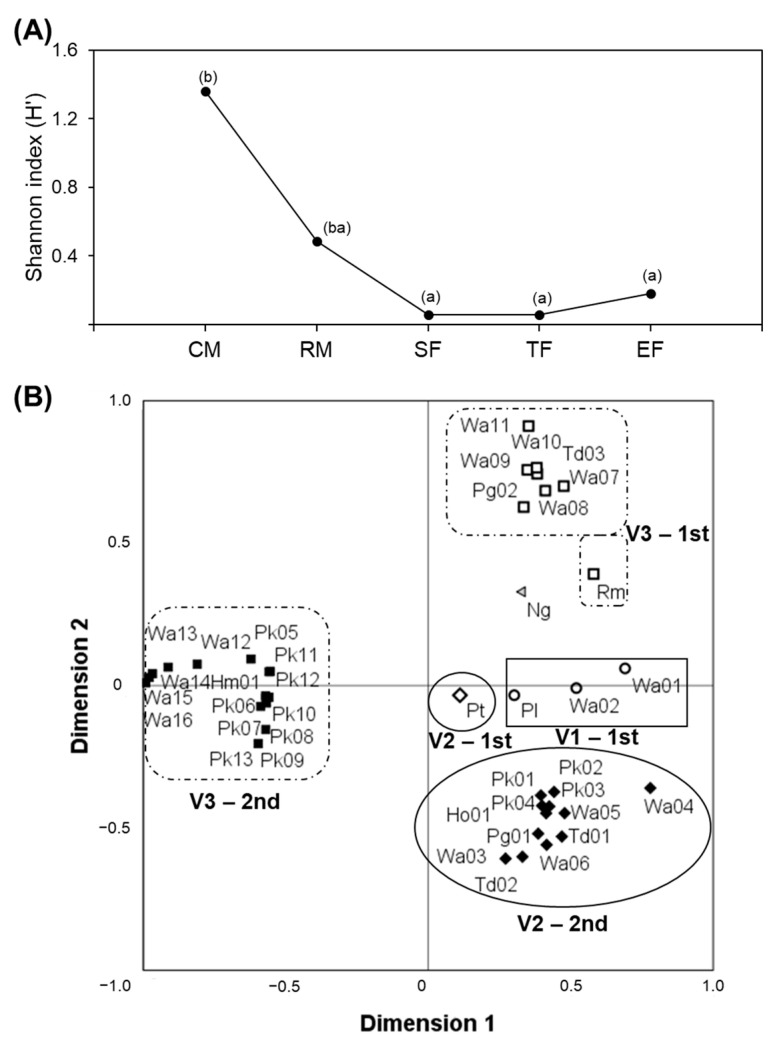
Diversity of yeast populations. (**A**) Yeast community α-diversity based on Shannon index according to fermentation stages. Different letters indicate significant differences among different stages of winemaking process. (**B**) Yeast community dissimilarity (β-diversity) represented by non-metric multidimensional scaling (NMDS). The groups of strains representative of each vineyard population were defined by different symbols: circles (V1), rhombuses (V2), or rectangles (V3), and they are delimitated by rectangles (V1), circles (V2), and squares (V3). The vintage populations (first and second) were distinguished for the colour of the symbols: white with black border (first); and fully black (second).

**Figure 4 foods-12-03644-f004:**
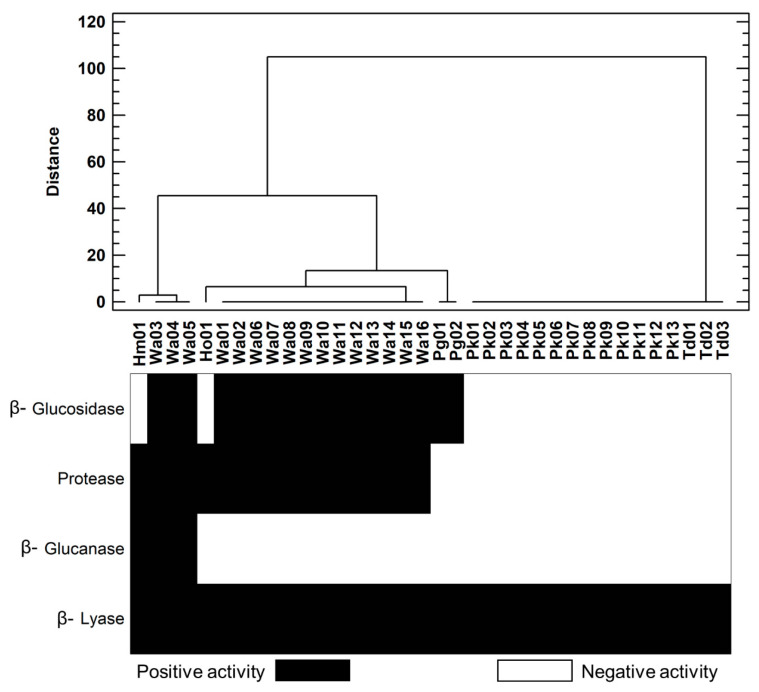
Determination of the enzymatic activities found in each of the non-*Saccharomyces* isolates represented by a heat map (positive and negative activity). The relationship of the isolates according to enzyme activities was represented by a dendrogram.

**Figure 5 foods-12-03644-f005:**
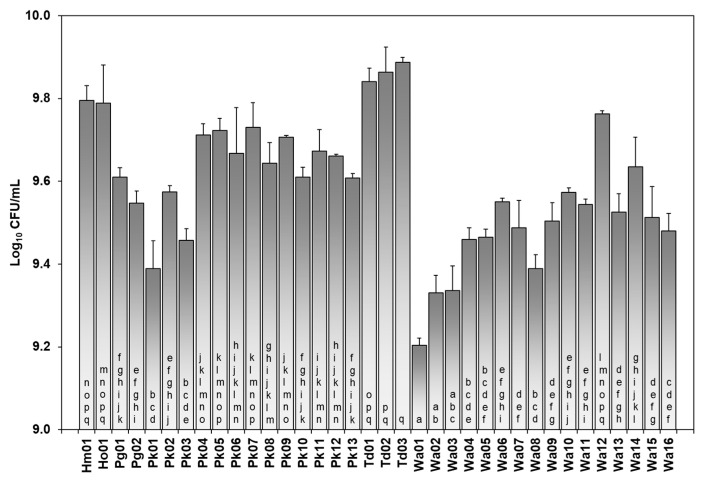
Quantification of β-lyase activity of the isolated yeasts determined as growth after 48 h (Log_10_CFU/mL). Different letters indicate significant differences in β-lyase activity among yeast isolates analysed.

**Figure 6 foods-12-03644-f006:**
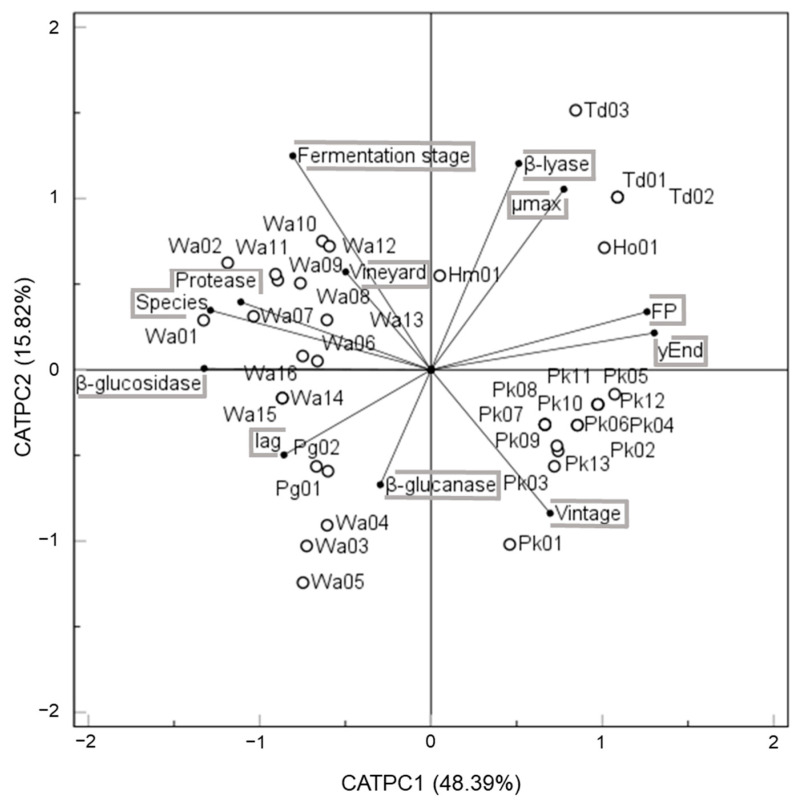
CATPCA biplot of first two principal components relating the oenological isolates of interest to ecological factors, fermentation characteristics, and enzymatic activities.

**Table 1 foods-12-03644-t001:** Kinetics parameters.

Isolate	µ_max_ (Days^−1^)	Lag (Days)	yEnd (g of Total CO_2_ Produced)	FP (Theoretical % vol. Ethanol)
*Hanseniaspora meyeri*				
Hm01	3.4 ± 0.8	-	39.7 ± 11.4	5.0 ± 1.4
*Hanseniaspora osmophila*				
Ho01	9.6 ± 2.0	-	86.9 ± 12.5	10.9 ± 1.6
*Pichia guilliermondii*				
Pg01	1.2 ± 0.1 ^a^	4.6 ± 0.2 ^a^	15.6 ± 1.6 ^a^	1.9 ± 0.2 ^a^
Pg02	1.4 ± 0.1 ^a^	3.9 ± 0.7 ^a^	18.7 ± 1.0 ^a^	2.3 ± 0.1 ^a^
*Pichia kudriavzevii*				
Pk01	1.8 ± 0.8 ^a^	1.7 ± 0.9 ^a^	34.3 ± 12.5 ^a^	4.3 ± 1.6 ^a^
Pk02	2.4 ± 0.5 ^a^	0.7 ± 0.3 ^a^	39.4 ± 12.5 ^a^	4.9 ± 1.6 ^a^
Pk03	2.8 ± 0.7 ^a^	0.2 ± 0.0 ^a^	47.4 ± 15.7 ^ab^	5.9 ± 2.0 ^ab^
Pk04	4.1 ± 1.3 ^a^	1.9 ± 1.1 ^a^	64.7 ± 5.6 ^ab^	8.1 ± 0.7 ^ab^
Pk05	3.2 ± 0.3 ^a^	1.0 ± 0.0 ^a^	62.2 ± 3.9 ^ab^	7.8 ± 0.5 ^ab^
Pk06	3.2 ± 0.7 ^a^	1.6 ± 0.0 ^a^	68.0 ± 1.0 ^ab^	8.5 ± 0.0 ^ab^
Pk07	2.6 ± 0.4 ^a^	-	45.9 ± 7.7 ^ab^	5.7 ± 1.0 ^ab^
Pk08	3.2 ± 0.4 ^a^	-	51.6 ± 9.3 ^ab^	6.4 ± 1.2 ^ab^
Pk09	2.7 ± 0.2 ^a^	-	51.3 ± 1.7 ^ab^	6.4 ± 0.2 ^ab^
Pk10	3.5 ± 1.5 ^a^	2.0 ± 0.3 ^a^	75.7 ± 4.2 ^b^	9.5 ± 0.5 ^b^
Pk11	3.1 ± 0.1 ^a^	-	54.6 ± 6.0 ^ab^	6.8 ± 0.8 ^ab^
Pk12	3.3 ± 0.2 ^a^	0.9 ± 0.0 ^a^	67.9 ± 3.8 ^ab^	8.5 ± 0.5 ^ab^
Pk13	3.6 ± 0.8 ^a^	2.8 ± 0.0 ^a^	63.5 ± 13.0 ^ab^	7.9 ± 1.6 ^ab^
*Torulaspora delbruckii*				
Td01	6.1 ± 0.4 ^a^	-	70.6 ± 8.7 ^a^	8.8 ± 1.1 ^a^
Td02	6.9 ± 1.3 ^a^	-	81.0 ± 8.3 ^a^	10.1 ± 1.0 ^a^
Td03	6.9 ± 0.9 ^a^	-	86.0 ± 10.6 ^a^	10.8 ± 1.3 ^a^
*Wickerhamomyces anomalus*				
Wa01	1.8 ± 0.1 ^a^	2.4 ± 0.1 ^a^	19.4 ± 2.0 ^ab^	2.4 ± 0.2 ^ab^
Wa02	2.2 ± 0.8 ^a^	2.7 ± 1.4 ^a^	19.3 ± 2.6 ^ab^	2.4 ± 0.3 ^ab^
Wa03	2. 0 ± 0.4 ^a^	2.4 ± 0.7 ^a^	17.4 ± 3.0 ^ab^	2.2 ± 0.4 ^ab^
Wa04	2.3 ± 0.3 ^a^	1.5 ± 1.0 ^a^	21.1 ± 5.8 ^ab^	2.6 ± 0.7 ^ab^
Wa05	1.8 ± 0.1 ^a^	1.5 ± 0.8 ^a^	14.6 ± 1.5 ^ab^	1.8 ± 0.2 ^ab^
Wa06	2.4 ± 0.2 ^a^	1.7 ± 0.6 ^a^	18.3 ± 1.1 ^ab^	2.3 ± 0.1 ^ab^
Wa07	2.7 ± 0.1 ^a^	2.2 ± 0.4 ^a^	17.7 ± 3.7 ^ab^	2.2 ± 0.5 ^ab^
Wa08	2.6 ± 0.2 ^a^	1.3 ± 1.0 ^a^	24.7 ± 3.6 ^b^	3.1 ± 0.5 ^b^
Wa09	2.1 ± 0.9 ^a^	2.0 ± 0.7 ^a^	14.6 ± 1.8 ^ab^	1.8 ± 0.2 ^ab^
Wa10	2.3 ± 0.7 ^a^	0.4 ± 0.2 ^a^	24.7 ± 4.7 ^b^	3.1 ± 0.6 ^b^
Wa11	2.0 ± 0.6 ^a^	1.8 ± 0.4 ^a^	17.5 ± 4.8 ^ab^	2.2 ± 0.6 ^ab^
Wa12	2.2 ± 0.7 ^a^	1.2 ± 1.7 ^a^	22.2 ± 7.0 ^ab^	2.8 ± 0.9 ^ab^
Wa13	2.0 ± 0.6 ^a^	0.6 ± 0.8 ^a^	21.6 ± 8.0 ^ab^	2.7 ± 1.0 ^ab^
Wa14	1.8 ± 0.1 ^a^	1.1 ± 1.0 ^a^	16.3 ± 3.8 ^ab^	2.0 ± 0.5 ^ab^
Wa15	1.5 ± 0.5 ^a^	1.5 ± 1.0 ^a^	11.2 ± 2.4 ^a^	1.4 ± 0.3 ^a^
Wa16	2.0 ± 0.1 ^a^	1.5 ± 0.8 ^a^	13.1 ± 1.4 ^ab^	1.6 ± 0.2 ^ab^

Kinetic was followed by CO_2_ production in microfermentations carried out by triplicate, and the curves were fitted to a sigmoid function using DMFit. Potential maximum rate, µ_max_ (days^−1^), lag phase, lag (days), and maximum CO_2_ production, yEnd (g of total CO_2_ produced), were estimated according to this software. Fermenting power, FP (theoretical % vol. ethanol), was calculated using yEnd data. Different letters in the same yeast species indicate significant differences among the isolates analysed (*p* < 0.05).

## Data Availability

Data are contained within the article or Appendix A.

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
