# Peer review of "Non-Saccharomyces Yeasts from Organic Vineyards as Spontaneous Fermentation Agents"

_foods, 2023, doi:10.3390/foods12193644_

Round 1

Reviewer 1 Report

The manuscript entitled “Non-Saccharomyces yeasts from organic vineyards as spontaneous fermentation agents” gives us useful information of non-saccharomyces species from Verdejo grapes, including the identification, metabolic characteristics, and the enzyme activities of several important enzymes. Meanwhile, as one of the most important properties of non-saccharomyces species is metabolically synthesis of aroma chemicals, the detection of some of the aroma chemicals should be offered. As the authors refer that the β-lyase is related to the synthesis of terpenes in the wine. The authors offered the data about β-lyase, while no data about terpenes.

All the non-saccharomyces species should be deposited in a public strain storage center so that other researchers could access such species only at research level.

The table legens is too simple.

The English language is qualified for publication after minor revision.

Reviewer 2 Report

Title: Non-Saccharomyces yeasts from organic vineyards as spontaneous fermentation agents

Overall: Good write up on the profiling of non-saccharomyces yeast from vineyard fermentation. Impressive and comprehensive new information were highlighted in this paper that would contribute to a better understanding of the role of these yeasts in wine fermentation. However, very minor suggestions are provided here to further improve the manuscript.

Abstract:          Need to be improved by including some other results such as enzyme activity, and fermentation kinetics, etc.

Introduction:          Good. Cover the relevant topics related to the research.

Materials and methods:          Good. Comprehensive.

Results and discussions:          Please include units for parameter inside Table 1 for clear understanding.

Conclusion:          Good

Reviewer 3 Report

The article shows the isolation of non-saccharomyces species from grape juice fermentations and provides confirmatory results. The novelty is related to the grape variety Verdejo used in the study. The results are in line with what would be expected from this type of experiment. The article is well written and the methods were clearly described.  The following issues are addressed:

1.       Main question

Authors should discuss the possibility of having damaged grapes during the fermentation. Indeed, winery ferments always include some rot grapes even if not visible. Furthermore, some of the isolated species are consistent with the presence of rot grapes.

Authors must refer to the work of Barata et al. concerning the microbial ecology of grapes and the influence of sour rot. The influence of rot is much more significant than the influence of any terroir effect, authors should quote the Alexandre (Microorganisms, 2020, 8(5), 787) critical review.

2.       Secondary questions

– In Material and methods indicate the harvest year of the two harvests and try to find if there was a rainier year that could justify the presence of rot and thus the increased species diversity.

– Indicate the volume of fermentation in the winery and if there was sulphite addition.

– Give former names for the species like Papiliotrema and Naganishia. It helps the reader to understand the technological significance of the isolates.

– Microfermentations in 50 mL autoclaved juice is far from reality. The results are preliminary.

– Enzymatic activity in culture media only serves as a first screening. Results are also preliminary.  

We advise the authors to present the limitations of the study section where these questions could be addressed.

Round 2

Reviewer 3 Report

Authors answered to the questions.